# Broad fault zones enable deep fluid transport and limit earthquake magnitudes

Konstantinos Leptokaropoulos [1,2] ✉, Catherine A. Rychert [1,3] ✉,
Nicholas Harmon[1,3], David Schlaphorst [4], Ingo Grevemeyer [5],
John-Michael Kendall [6] & Satish C. Singh [7]

Constraining the controlling factors of fault rupture is fundamentally important. Fluids influence earthquake locations and magnitudes, although the exact pathways through the lithosphere are not well-known. Ocean transform faults are ideal for studying faults and fluid pathways given their relative simplicity. We analyse seismicity recorded by the Passive Imaging of the Lithosphere-Asthenosphere Boundary (PI-LAB) experiment, centred around the Chain Fracture Zone. We find earthquakes beneath morphological transpressional features occur deeper than the brittle-ductile transition predicted by simple thermal models, but elsewhere occur shallower. These features are characterised by multiple parallel fault segments and step overs, higher proportions of smaller events, gaps in large historical earthquakes, and seismic velocity structures consistent with hydrothermal alteration. Therefore, broader fault damage zones preferentially facilitate fluid transport. This cools the mantle and reduces the potential for large earthquakes at localized barriers that divide the transform into shorter asperity regions, limiting earthquake magnitudes on the transform.

The role of fluids is fundamental to understanding earthquake rupture location, size and style[1–4]. Ocean transform faults are ideal places to study the factors controlling seismicity, since they are relatively simple in comparison to their hazardous, continental counterparts[5–7]. The composition of the oceanic lithosphere is thought to be relatively homogeneous, and the associated fault structures are typically simple i.e., nearly linear and vertical with deformation localized in a narrow, 1–15 km wide, zone[8]. Plate velocities are also relatively well-known based on magnetic lineations and/or plate motion reconstructions[9]. Finally, studies of teleseismic earthquake locations[10] and earthquake rupture experiments[11] show that the depth limit of the region that can potentially slip seismically occurs at the brittle ductile transition, i.e., near the 600 °C isotherm, which is well predicted by thermal models.

Despite this relative simplicity, several observational constraints suggest there is still much to learn regarding the factors that control seismogenic processes. For instance, the amount of moment released seismically on transform fault systems is smaller than that predicted by plate motions and simple thermal models, ranging from 0 to 83% globally with a mean of 18%[7]. In addition, oceanic transforms typically do not generate earthquakes much larger than M = 7, i.e., much smaller than the potential size of rupturing the entire transform length[5]. For instance, at the Chain Transform Fault the magnitude of rupturing the entire transform is predicted to be M > 8.0[12] based on empirical relationships. Finally, earthquakes have been located deeper than the predicted depth of the brittle-ductile transition from thermal models at the Gofar, Blanco, and Romanche Transform Faults[13–15].

Oceanic transforms are thought to be regions where large amounts of fluids enter the lithosphere[4]. Specifically, fluids within the fault zone directly affect the pore pressure (i.e., the effective normal stress), potentially advancing the time-to-failure[1,2,16]. This process can

[1]Ocean and Earth Science, University of Southampton, Southampton, UK. [2]The MathWorks, Cambridge, UK. [3]Geology and Geophysics, Woods Hole Oceanographic Institution, Woods Hole, USA. [4]Instituto Dom Luiz (IDL), Faculdade de Ciências, Universidade de Lisboa, Lisbon, Portugal. [5]GEOMAR Helmholtz Centre of Ocean Research Kiel, RD4—Marine Geodynamics, Kiel, Germany. [6]School of Earth Sciences, University of Oxford, Oxford, UK. [7]Université de Paris, Institut de Physique du Globe de Paris, CNRS, Paris, France. ✉e-mail: K.Leptokaropoulos@soton.ac.uk; catherine.rychert@whoi.edu

reduce the likelihood of large earthquake rupture[17,18]. Fluids also alter the surrounding rocks, forming serpentinites and other hydrous mineral phases that coat the fault interface and are characterized by weaker rheological and frictional behavior than their peridotitic and mafic protoliths[19]. In addition, hydration decreases the strength of olivine, which results in higher strain rates, by up to an order of magnitude, potentially facilitating aseismic creep and/or micro-seismicity rather than large-scale ruptures[20]. Hydrothermal circulation may also cool the lithosphere extending the brittle ductile transition and resulting in deeper seismicity[13,14,21], and the degree to which this occurs may vary along the fault[22]. Geodynamic models suggest that dilatancy mechanisms, whereby fluids become more viscous under pressure may be in effect periodically[23,24]. Fluids thus play a primary role. However, their exact pathways and relationship with observed earthquakes are not well-known.

We present the seismicity recorded by the deployment of 39 ocean bottom seismometers (OBS) from March 2016–2017 surrounding the Chain Fracture zone as part of the Passive Imaging of the Lithosphere-Asthenosphere Boundary (PI-LAB) Experiment and the Experiment to Unearth the Rheological Lithosphere-Asthenosphere Boundary (EURO-LAB)[25–30]. The seafloor of Chain Transform Fault varies in age from 0 – 20 Myr old across the rupture zone. The active fault zone has a width of up to ~15 km, stretching over 300 km in roughly the east-west direction between two spreading segments of the Mid-Atlantic Ridge, and is slipping at a rate of ~30 mm/yr.

The Chain Transform Fault contains four lozenge-shaped topographic highs and shows evidence for variability in gravity anomalies and inferred crustal structure along the length of the transform fault[25] (Fig. 1a, b and 2). This morphology results from the exhumation of crustal blocks due to transpressional stresses, and these blocks are referred to as positive flower structures (Methods). The eastern section is dominated by the largest flower structure (ELFS; Figs. 1a and 2b), the central section has no flower structures, and the western section is characterized by three small flower structures: western

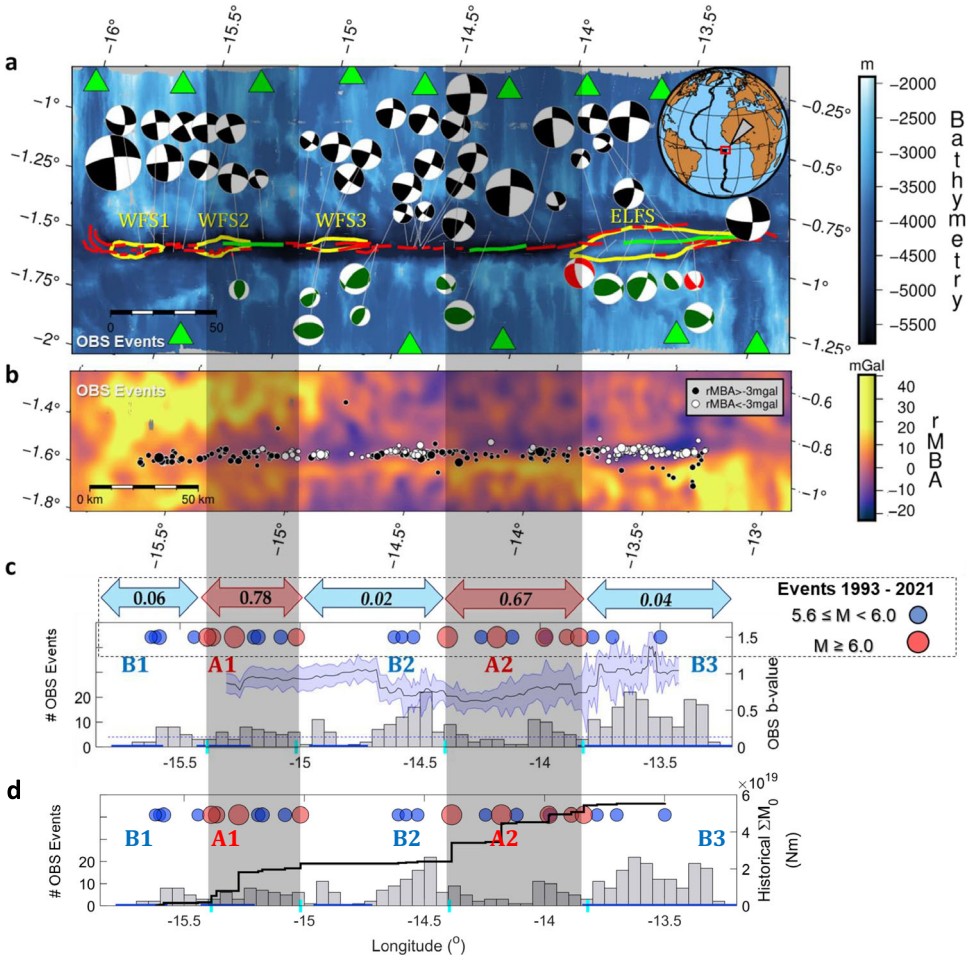

**Fig. 1 | Seismicity along Chain transform fault. a** Bathymetric map of the study area with the four topographic highs, i.e., the western flower structures (WSF) and the Eastern Large Flower Structure (ELFS), all represented as yellow curves. The red and green curves show the fault scarps, with the latter highlighting the longest of them (L > 30 km). We use an oblique Mercator projection so that the transform appears horizontal, which is the reason for angled ticks marking longitude and latitude. High quality focal mechanisms calculated from the Ocean Bottom Seismometer (OBS) data projected as lower hemispheres. Events with dominant reverse and normal components are indicated by green and red beachballs, respectively. Inset map shows the study region (red box). OBS stations are shown by green triangles. Horizontal resolution of the bathymetry is 100 m and vertical resolution is ~10 s m. **b** Seismicity epicenters in the OBS data superimposed over the residual mantle Bouguer anomaly (rMBA) values in the study area. Black and

white dots indicate events along the transform valley, located on high rMBA (> −3 mgal) and low rMBA (<−3 mgal), respectively. **c** Histogram with the number of OBS earthquake events per 0.05° longitudinal bins. The blue and red circles show the historical events with $M_W ≥ 5.6$ and $M_W ≥ 6.0$, respectively[31]. The number in the arrows show the seismically released moment fraction for these segments (Methods). The black curve shows along strike variation of b-value in Chain, considering equally sized event windows of 75 OBS events each, advancing by 1 event and the standard error of b-value is defined by the purple area. The horizontal blue lines at the x-axis indicate the location of the positive flower structures. The shaded areas correspond to asperities (A1 and A2) which separate the barriers (B1, B2, B3). **d** Along-strike cumulative seismic moment of the historical events (black line). The other panel features as in **c**.

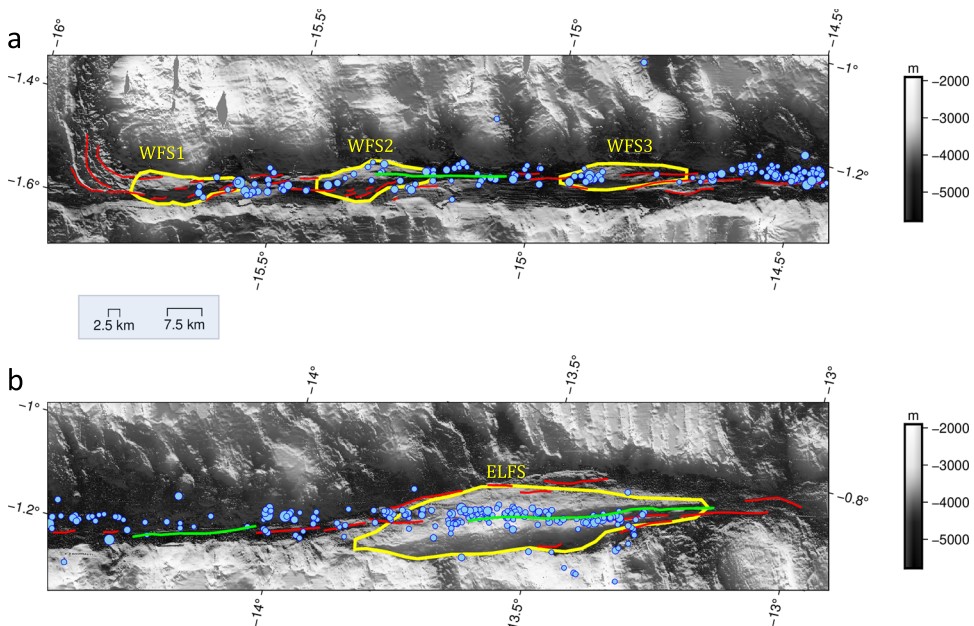

**Fig. 2 | Flower structures and fault scarps. a** Western half of Chain transform fault and b) eastern half of Chain transform fault. The red and green curves show the fault scarps, with latter highlighting the longest of them (L > 30 km). The yellow curves denote the positive flower structures (WFS 1–3 Western Flower Structures, ELFS Eastern Large Flower Structure). The bathymetry is shown in gray scale. The blue circles indicate the location of the events above $M_C$ = 2.3 that were considered in the study. The median (50% quantile) and 95% quantile of the lateral uncertainty are shown in the inset box, and correspond to 2.5 km and 7.5 km, respectively.

flower structures 1, 2 and 3 (WFS1, WFS2, and WFS3; Figs. 1a and 2a) from west to east.

The Chain Transform Fault is also characterized by several fault scarps. There are three relatively long segments (green lines, Fig. 2): one 30 km long scarp intersecting WFS2 and extending 20 km to the east, one 30 km scarp roughly mid-way between WFS3 and ELFS, and one 50 km long scarp located completely within ELFS. The longer fault segments are separated by much shorter fault scarps, typically 2–10 km in length. Multiple parallel and overlapping fault strands, arranged as en-echelon structures occur primarily within and nearby the flower structures, with relatively straight, singular scarps observed in the remaining sections of the transform fault. There could be additional fault structures beneath the resolution of our bathymetry, but there is no reason to believe that they would be different or more important than the large-scale structures on which we have based our arguments.

Here we show that earthquakes extend to deeper depths beneath the transpressional features characterized by broad damage zones with parallel fault segments and step overs. These regions are also coincident with gaps in large historical seismicity, slow sub-Moho P-wave velocities suggestive of hydrothermal alteration, deeper faster S-wave velocities that are consistent with enhanced hydrothermal cooling of the lithosphere, and higher b-values, indicative of greater numbers of smaller earthquakes. This implies that these broad fault damage zones preferentially facilitate fluid circulation into the lithosphere. The fluid circulation cools the mantle to deeper depths and increases the depth of the brittle-ductile transition. Fluid alteration also weakens the mantle, resulting in smaller earthquakes. These weakened zones act as barriers that segment the fault and effectively limit the potential for large earthquakes on the transform.

## Results
### Historical seismicity
25 $M_W$ ≥ 5.6 historical earthquakes along the Chain Transform Fault are reported in the relocated catalog[31] which extends back to 1993 (Fig. 1c). Empirical relationships[12] suggest a fault rupture length of ~33 km for the largest ($M_W$ = 6.6) recorded earthquakes, in

agreement with the scale of two of the longest fault scarps[25] that are located on the eastern side of WFS2 and between WFS3 and ELFS (~30 km) (Figs. 1 and 2). No large earthquakes have been recorded from the longest, 50 km, fault length located within ELFS since 1993 (green lines, Figs. 1 and 2)[31]. Assuming this rough segmentation scale of 30–50 km, the Chain Transform can be divided into 2 asperity regions (A1, A2) alternating with three barrier regions (B1, B2, B3) where no large ($M_W$ ≥ 6.0) earthquakes have been recorded but are rather characterized by aseismic slip and/or seismicity swarms (Fig. 1, Supplementary information Fig. 1). B1 and B3 are located near the ends of the transform. Transform sections near the ridge, e.g., within 20 km[31] do not typically host the largest earthquakes given the rapidly changing stress regime[32]. However, B1 and B3 extend to much further distances from the ridge (>50 km) and include substantial length scales within regions predicted to be characterized by more uniform stress regimes[32]. Therefore, seismicity characteristics are not necessarily only a function of proximity to the ridge in these locations. The three barrier regions are co-located with the location of flower structures (WFS1, WFS3, and ELFS), whereas the asperity regions are co-located with WFS2 and the central region of the transform where no flower structures are present. Two of the longest mapped fault scarps also coincide with the asperity regions, occurring on the eastern side of A1 and in A2 (Fig. 1). The amount of historical moment released seismically varies considerably along the fault (Methods, Fig. 1c, d). Beneath barriers B1, B2, and B3 the moment released seismically is small in comparison to the expected potential (2–6 %). Beneath asperities A1 and A2 moment release is much closer to the predicted potential (78% and 67%, respectively).

### Local seismicity recorded by the OBS
We detected 626 local events (370 above the magnitude of completeness, $M_C$ = 2.3) with locatable epicenters[33,34] (Fig. 1; Methods). The number of events per km along strike was highest (2.5 events/km) in segment B3 (Fig. 1b, c). There were 0.54 and 1.36 events/km in B1 and B2, respectively, and 0.98 and 0.82 events/km in A1 and A2, respectively. Depths were sufficiently constrained for 89 events (Methods).

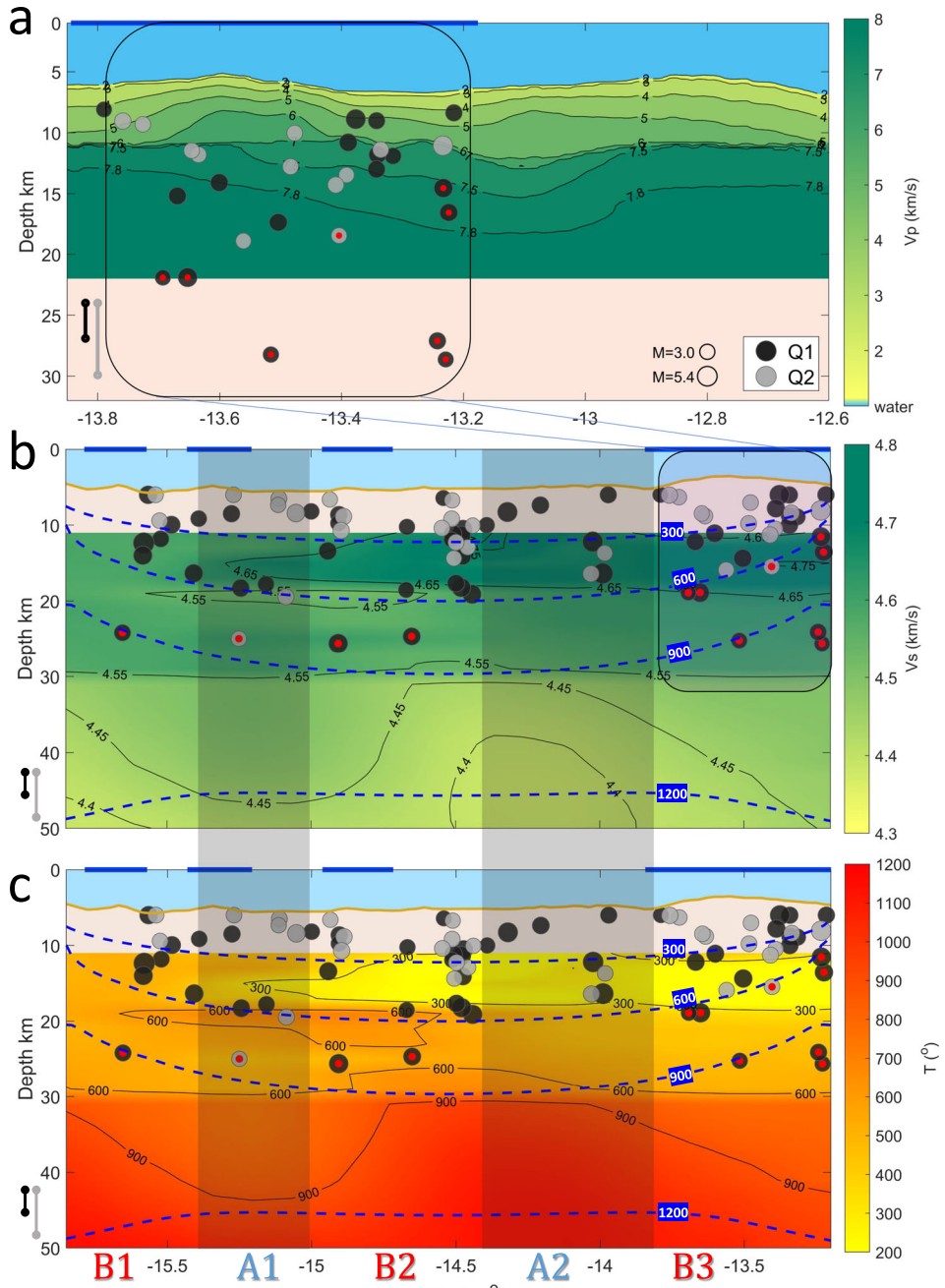

**Fig. 3 | Cross section of Chain transform fault.** Vertical distribution of 89 events determined by moment tensor inversion (circles). Panel **a** shows the eastern part of Chain, focusing on the Eastern Large Flower Structure (ELFS), defined by the rectangle in **b**. The depth determination quality (Q1 and Q2) is indicated as dark and light circles, respectively. Mean vertical uncertainties are 3 km for Q1 events (black vertical error bars) and 6 km for Q2 events (gray vertical error bars). The events located deeper than the 600 °C isotherm from simple thermal models (dashed blue lines in panels **b** and **c**) are highlighted with a red dot. The horizontal blue lines denote the location of the transpressional flower structures, whereas the vertical shaded bars indicate the asperities (A1, A2), separated by barriers (B1, B2, B3).

Depths are below sea level. **a** P-wave refraction model from the active source survey at the eastern part of Chain, beneath the ELFS (see Methods; Supplementary Data 1). **b** S-wave wave velocity model[35]. The dashed blue isolines (and numbers in blue boxes) indicate the thermal structure as determined by the prediction from a half-space cooling model for a transform region, assuming a half-spreading rate of 16 mm/yr. **c** Thermal structure predicted by the S-wave wave velocity model shown in **b** using the relationships of Stixrude and Lithgow-Bertelloni[47]. For additional approaches and assumptions for the thermal calculation please see Methods, Supplementary Fig. 2.

These events were distributed from just below seafloor (bsf) to ~22 km bsf, mostly at <15 km depth (82 events). 7 events were located between 19 and 22 km bsf, well below the depth of the predicted brittle-ductile transition based on the 600 °C isotherm from simple thermal models (Fig. 3, Supplementary Fig. 2). The deepest events were located directly beneath the WFSs and the ELFS. For 47 events it was possible to determine robust focal mechanisms (Fig. 1a, Methods). Most of those

earthquakes (34) occurred with predominantly strike-slip focal mechanisms. However, there were 13 earthquakes with a dominant vertical component (45° <|rake|<135°), 2 of which were normal and 11 of which were reverse, and 11 of these are located directly beneath or within 5 km of flower structures (Fig. 1a, Methods). Stress inversion results suggest that simultaneous activation of both strike-slip and reverse faults is possible (Methods, Supplementary Fig. 3). Elevated

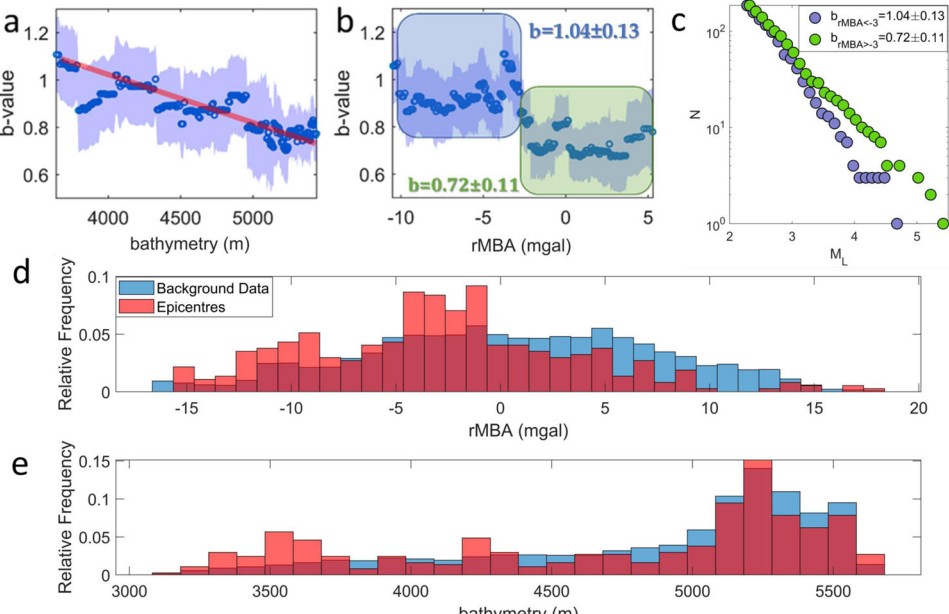

**Fig. 4 | OBS seismicity distribution with geophysical properties.** b-value fluctuation with bathymetry (**a**) and residual Mantle Bouguer Anomaly (rMBA) (**b**). The blue dots show the b-value estimated by the repeated median technique for 100-event windows, plotted at the center of each window (Methods). The windows are shifted by 1 event after each calculation. The red line in **a** shows the average linear trend. The shaded area shows the standard error of the b-values. The blue and green boxes in **b** show the b-values for rMBA < −3 mgal and rMBA > −3 mgal,

respectively. **c** Magnitude distribution of the events that occurred at areas with rMBA < −3mgal (blue circles) and rMBA > −3 mgal (green circles). Both groups have equal number of events (185 events each). All standard errors are derived by 1,000,000 bootstrap resamplings. Relative frequency of rMBA (**d**) and bathymetry (**e**) corresponding to the epicenters of our complete dataset events (red bars) and to the background sample (blue bars; see Methods).

b-values are found in regions of shallow bathymetry and low rMBA, i.e., characteristics common to the flower structures (Figs. 1 and 4, Methods; Supplementary Movie 1). The length scale of heterogeneity in seismicity is 10 s of km, which is much greater than the scale of bathymetric resolution (100 m), allowing us to make meaningful comparisons.

### Seismic velocity and thermal structure

We interpret our result considering two seismic velocity models. We present a P-wave refraction model to enhance our interpretation of shallow (<20 km depth below sea level) depths beneath the ELFS (Fig. 3a, Methods, Supplementary Fig. 4, Supplementary Fig. 5). We also compare our result to thermal structure predicted from the S-wave model derived from local Rayleigh and ambient noise tomography (Methods, Fig. 3c)[35]. We preferentially interpret the P-wave model at shallow depths beneath the ELFS, but also consider the S-wave model for deeper velocities and coverage along the entire transform (Methods).

Seismic velocities can be influenced by a variety of factors such as composition including degree of alteration/hydration, pore fluid, or temperature. Typically, the bulk composition of depleted mantle is expected to be relatively homogeneous, but even large variations in depletion would yield very minor amounts of seismic velocity anomalies[36]. Hydrothermal alteration and serpentinization are expected to reduce seismic velocities. However, the S-wave velocities we observe in the lithosphere are relatively high, 4.55 km/s on average, and inconsistent with these factors[37]. Also, porosity effects on S-wave speeds are not likely to be important at the mantle depths where surface waves have sensitivity (>~20 km depth)[38]. Therefore, we proceed interpreting the S-wave velocities in terms of temperature. Observed slow P-wave velocities could be explained by thickened crust, enhanced porosity, and/or hydrothermal alteration, which we will discuss in greater detail in the next section.

## Discussion

Beneath barrier B3/ELFS there are several observations that support the influence of greater damage, porosity, and/or fluids and associated hydrothermal circulation and alteration. Clear Moho reflections are visible across the region from the active source experiment, and these arrivals support relatively constant crustal thickness at 11 km beneath the sea surface (Supplementary Data 1; Supplementary Fig. 6); however, anomalously slow P-wave velocities in the subcrustal lithosphere are present down to at least 15 km depth beneath the sea surface, which are consistent with fluids and damaged, porous, and/or altered low density mantle material. In addition, the observed low rMBA also requires lower density mantle material, especially given that the crust is not particularly thick, which is consistent with this interpretation. Alteration is likely more important at mantle depths where lithostatic stress increases[38]. Fluids may also reduce the frictional properties of the transform fault beneath ELFS and explain why the longest (50 km) fault scarp in the region (Fig. 1, green line on ELFS, Fig. 2) is not associated with the largest earthquakes, but instead associated with elevated b-values (1.1–1.2) with very low fractions of seismically released moment (4%). Fluid infiltration into the crust and very shallow lithospheric mantle would also result in cooler temperatures and associated higher seismic velocities at greater depths. Indeed, surface wave-derived S-wave velocities indicate higher velocities on average over the shallow mantle (down to 40 km depth) beneath the flower structures (Fig. 3b). The predicted thermal contour depths (e.g., 600 °C) and shapes (e.g., 900 °C) are much different than the predictions for a simple conductive cooling thermal model, which suggests cooling to greater depths in general in these regions, accounting for the depth sensitivity of surface wave inversion (Methods). Enhanced hydrothermal circulation and cooling of the lithosphere may also explain the deeper seismicity beneath B3/ELFS due to a deepened brittle-ductile transition.

Observations in asperity A2 are much different than B3/ELFS, likely more consistent with lower degrees of hydrothermal alteration.

b-values are lower, and there have been several M ≥ 6.0 historical earthquakes (Fig. 1c), both consistent with a relatively strong, velocity weakening fault segment, and the lack of small earthquakes caused by fluid migration. The moderate rMBA is consistent with a normal crustal thickness and does not require high degrees of alteration. There is a lack of seismicity at >20 km depth, i.e., the depth of the 600 °C from simple thermal models, consistent with a more typical depth to the brittle ductile transition. In addition, the predicted temperatures from the S-wave velocity model averaged over depth, given the sensitivity of surface waves (Methods), are hotter than surrounding regions which could be consistent with less hydrothermal cooling in this region (Fig. 3b). Resolution testing suggests that the observed variations in the seismic velocity model from which the predicted temperatures are derived along the transform fault are robust (Methods).

The WFSs observations share some similarities to those of ELFS, but also some differences. Similarities include high topography, broad deformation zones, and relatively high S-wave velocities and likely cooler temperatures on average which could be consistent with larger degrees of hydrothermal circulation in the region. The rMBA in the WFSs region changes from negative to positive from east to west. The higher values are likely caused by normal to thinner crustal thickness, which likely dominates the gravity signature[25]. The large historical earthquakes beneath WFS2 also distinguish it from the other flower structures. One explanation may be that the long fault segment in the region extends 20 km east of the flower structure and may behave more similarly to the long fault in section A2. Another explanation could be that there are temporal variations in rheological behavior as water is delivered and transported through the system. Episodic fluid delivery and transport and multi-mode rupture models have been previously proposed to explain observed seismicity patterns[5,7,39] and have also been predicted by geodynamic modeling[24] and inferred from samples from exhumed oceanic transform faults[40].

A few other studies have reported transform fault seismicity at deeper depths than the 600 °C isotherm[13–15]. Geodynamic models predict a deepened 600 °C isotherm if there is hydrothermal circulation, and this is true particularly near the ends of the transform[21]. However, seismicity has also been reported at deeper depths than predictions including hydrothermal circulation[21]. Therefore, it has been suggested that this can be explained by higher strain rates at transform faults[41]. Alternatively, different rheological flow laws may be in effect, and brittle and ductile deformation may occur over a broad range of temperatures (300–1000 °C) and, therefore, a broad range of depths, owing to variable seawater infiltration and grain sizes along the fault[22]. Our result does not preclude the possibility that either or both scenarios are in effect in some locations globally. However, what is different about our work is that our S-wave velocity model offers an independent constraint that is not often available. It shows that different flow laws and/or high strain rates are not required at Chain, since seismicity occurs shallower than the 600 °C isotherm derived from the S-wave velocity model. In other words, our S-wave velocity model demonstrates that enhanced damage results in increased hydrothermal circulation, which has cooled our region more efficiently and to deeper depths beneath the flower structures. This also likely occurs elsewhere, particularly in transpressional environments, for example, St. Paul and Owen Transform Faults[42,43].

The Chain Transform Fault provides a view into how segmentation on oceanic transform faults can develop, which may in turn explain why oceanic transform faults do not generate larger earthquakes given their fault lengths. Although previous studies have suggested that transforms may be segmented, and it has been hypothesized that variable water infiltration into the lithosphere plays a role, our study provides new constraints on the Earth properties that can result in such phenomena. Multiple overlapping faults located on the flower structures are associated with broader damage zones, which provide a wider pathway for fluids into the crust and uppermost

mantle. The presence of fluids causes a series of mechanical and chemical effects, potentially increasing the pore pressure, reducing the effective normal stresses, lubricating the fault, and/or creating weaker regions of altered material. The hydrothermal circulation that occurs at shallow depths also cools the underlying mantle deepening the brittle-ductile transition. However, the damaged/altered crust and upper mantle are weaker, permitting only low magnitude earthquakes and/or aseismic accommodation of tectonic strain accumulation in these barrier regions beneath the flower structures. The barrier regions separate stronger, locked regions (asperities) beneath singular, linear fault strands with narrower damage zones, where hydrothermal circulation is lower, and the brittle ductile transition and observed earthquake depths are shallower. Although the faults in asperity regions are stronger and larger earthquakes occur there, the barrier regions effectively segment the Transform Fault. This in turn limits the maximum magnitude that could occur on the transform.

## Methods

### Origin of the lozenge-shaped topographic highs

One possibility is that these features represent intra-transform volcanism[44]. However, the observed low backscatter intensity of these structures[25] does not support recent volcanism. A more likely scenario is that these features represent crustal blocks exhumed via transpressional stresses underlain by altered mantle, i.e., features typically referred to as positive flower structures. This is also supported by the 11 reverse fault mechanisms recorded by the OBS along the transform (Fig. 1). There were only 2 events characterized by a dominantly normal focal mechanism, potentially explained by flank collapses. Similar morphological features are frequently observed and fault plane locations are sometimes available to substantiate a transpressional origin. This is supported by the highly tectonised mafic and ultra-mafic compositions of rocks from features with similar morphology and gravity signatures at the nearby St. Paul transform fault that are also thought to have a transpressional origin[42].

### Seismic catalog and focal mechanisms

The analysis of the broadband ocean bottom seismic deployment revealed 972 events, 812 of which are located within the area surrounded by the OBS network[33], which belongs to the Chain transform fault and the adjacent ridge spreading centers (Fig. 1b). The location of the events is performed by NonLinLoc software[45] using a 1-D velocity model of the Chain transform fault region based on CRUST1.0[46] and earthquake arrival times[47]. In order to focus on the transform fault seismicity, we discard the events that unequivocally occurred at the ridge segments and at the inside corners after considering their location and focal mechanisms (north-south striking, normal faulting events). This selection leaves 626 events along the Chain transform valley, within a local magnitude ($M_L$) range between 1.1 and 5.4. We further remove events below the completeness threshold $M_C = 2.3$ (see below), resulting to a dataset of 370 events along the transform valley, which is used for our analysis.

The epicentral coordinates of these events are sufficiently constrained with median lateral uncertainties of 2.5 km. Vertical uncertainties are larger (median ~18 km) and several events are above 20 km. For this reason, the hypocentre depths as well as focal mechanisms of well-recorded events are re-evaluated using the Grond software[48], which carries out a Bayesian-bootstrap time-domain deviatoric moment tensor inversion. We calculated 119 focal mechanism solutions. For the stress inversion we select the 47 focal mechanisms classified as having good-fits (Fig. 1). We include 42 additional events (89 events total) with sufficiently well-determined depth (mean vertical error of 6 km).

We supplement our seismicity data recorded by the OBS with historical earthquakes from the Global Centroid Moment Tensor catalog[49]. Due to the relatively large epicentral uncertainties, we only

consider the strong events ($M_W \geq 5.6$) after 1993 relocated by Shi et al. [31].

## OBS seismicity with respect to bathymetry and rMBA

Here, we investigate whether there is preference for seismic events in the local OBS data occurring at sites with particular rMBA/ bathymetry values. In doing so, we compare the rMBA/bathymetry values at the epicentral location of events with the background rMBA/bathymetry data. Background data are defined as all the rMBA/bathymetry values calculated within a 15 km wide zone, centered on the transform valley axis and bounded in the along-strike direction by our seismicity data (roughly between the ridge spreading centers). In such way, we consider only the values corresponding to the active deformation zone along the transform valley, where the vast majority of seismicity in the local OBS data occurs, discarding the essentially aseismic areas elsewhere. If the earthquake epicenters show no preference for particular parameter values, then the red and blue histograms in Fig. 4d and Fig. 4e should be similar. We see, however, that there is a clear preference of events occurring at negative rMBA values and an event deficit for positive rMBA (Fig. 4d). There is also a preference of events occurring at very shallow ocean depths (~<3700 m, Fig. 4e), i.e., directly beneath the ELFS. We perform two statistical tests, to quantify the difference between the two distributions. The Wilcoxon rank sum test is used to test the null hypothesis that the data in the two sets (i.e., epicenters and background data) are drawn from distributions with equal medians. In addition, the two sample Kolmogorov-Smirnov test was applied to test the hypothesis that the data in the two sets (i.e., epicenters and background data) are drawn from the same continuous distributions. Both Wilcoxon rank sum test and Kolmogorov-Smirnov test reject the corresponding null hypotheses: at $p < 10^{-6}$ for rMBA and at $p < 10^{-4}$ for bathymetry.

## Magnitude distribution

We investigate the seismicity magnitude distribution dependence on bathymetry and rMBA, after assigning a bathymetry and rMBA value to each earthquake by nearest neighbor interpolation. Then the events are sorted by bathymetry and rMBA values in an ascending order and the b-values from the corresponding event magnitudes are calculated, together with standard error. Calculations are performed for overlapping 75-event windows, sliding by 1 event (different windows of 50 to 100 events were also tested, as shown in Supplementary Movie 1).

The exponentiality of magnitude distribution is investigated by the Anderson-Darling Test[50,51] and further established by goodness of fit test[52,53], applying the Maximum Likelihood Estimation (MLE) of b-value[54]. Both techniques suggest that the magnitude distribution can be sufficiently modeled by an exponential distribution for $M_L \geq 2.3$. A total of 370 events with $M_L \geq 2.3$ comprise the complete data for the study area (circles, Figs. 1b and 2), which roughly corresponds to an average rate of ~1 event/day. However, the MLE returns considerably unstable b-values, even well above $M_C$, fluctuating between 0.76 and 0.89 for $2.3 \leq M_L \leq 3.0$ (Supplementary information Table 1, Supplementary Fig. 7). This is incompatible with the unique b-value predicted by the GR law and may lead to artifacts and misinterpretation of the derived outcome. For this reason, we apply alternative methods which are more suitable for small datasets and are less sensitive to $M_C$ selection[55,56]. The comparison of the results from the aforementioned methods, assuming different $M_C$ are shown in Supplementary Fig. 7. We choose to estimate b-values by applying the repeated medians[57] (RM), a non-parametric regression technique, which provides more robust results for small data sets, since it is highly resistant to observational uncertainties and outliers[55]. According to the RM approach, we consider n magnitude intervals between two points, i and j, having $N_i$ and $N_j$ events, respectively, and calculate n-1 slopes (b-values)

between these points as:

$$b_{ij} = \frac{\log_{10}(N_j) - \log_{10}(N_i)}{M_j - M_i} \quad (1)$$

With $M_j \neq M_i$. For each point, i, we calculate the median slopes, thus n median values. Then, the $b^{RM}$ is estimated as the median of the n median values:

$b^{RM} = -\text{median}(\text{median}(b_{ij}))$

The standard errors, SE, of the corresponding $b^{RM}$ are derived by bootstrap resampling process as described in Amorèse et al. [55]. The significance of the b-value difference for two datasets, e.g. A and B, can be estimated by a bootstrap t-test, with the t statistic defined as:

$$t = \frac{|b_A^{RM} - b_B^{RM}|}{\sqrt{SE_A^2 + SE_B^2}} \quad (2)$$

A $t = 1.96$ corresponds to a significance at 0.05 level. This approach returns $b = 0.83 \pm 0.09$, with this value being insensitive to the applied magnitude cut-off ($b = 0.82–0.84$ for $2.0 \leq M \leq 3.0$) as shown in Supplementary Fig. 7 and Supplementary Table 1. We therefore apply the RM technique for all b-value calculations in this study, since it provides the most consistent results, regardless the $M_C$ choice.

The b-values are highest beneath the ELFS (1.1–1.2), they are lowest (<0.8) in the central parts of the fault (-14.7°E–13.8°E) and intermediate (0.9–1.1) in the western area (Supplementary Movie 1). Shallower water depths generally correspond to higher b-values in comparison to the deeper bathymetry areas, where b-values gradually decline with increasing depth at -0.15 units/km (Fig. 4a). The apparent cyclic relationship between the b-values and bathymetry is an artefact related to the low number of events in each calculation (100). b-values calculated using a small number of data are more sensitive to the larger magnitude events, which coincide with the x-axis locations where jumps in b-value are apparent (Fig. 4a). There are higher b-values ($b = 1.04 \pm 0.13$) in regions of low rMBA (<−3 mgal) than regions of high rMBA (>−3 mgal) where $b = 0.72 \pm 0.11$ (Fig. 4b, c) with the difference between these two b-values being statistically significant at 0.05 level, indicated by the calculated t statistic equal to 1.96. Events also occur preferentially in areas with negative rMBA (Fig. 4d). These areas are mainly located in the regions of shallowest bathymetry (<3700 m), i.e., beneath the ELFS (Fig. 1; Fig. 4e).

## Seismic vs predicted moment

We compare the seismic moment release, $M_O$, in each asperity (A1, A2) and barrier (B1, B2, B3) using the formula[58] $M_O = 10^{1.5M_W + 9.105}$, where $M_O$ is given in N • m. We express the results in terms of percentage ($\alpha$) of seismically released moment ($\Sigma M_{Os}$) over the potential moment release assuming full seismic coupling ($\Sigma M_{0p}$), i.e., $\alpha = \Sigma M_{Os}/\Sigma M_{0p}$. In doing so, we estimate $\Sigma M_{0p}$ by summing the seismic moment of all $M_W \geq 5.6$ events occurred since 1993 in each patch (barrier or asperity). We then estimate $M_{0p}$ as[5]:

$$M_{0p} = GLws\Delta T \quad (3)$$

where, G is the shear modulus [40–60 GPa], L is the patch length, w, is the patch width from sea floor to the 600 °C isotherm, s is the slip rate [28–33 mm/yr] and $\Delta T$ is the time period duration since 1993, equal to 28 years. Our results range from $\alpha = 0.25$ to $\alpha = 0.46$ with an average value of $\alpha = 0.32$ (for $G = 50$ GPa and $s = 32$ mm/yr). This range agrees well with previous findings ($\alpha = 0.33$) for the entire Chain Transform[7]. However, we find the proportion of seismically released moment varies along the length of the transform from 0.02 to 0.06 in barrier

regions to 0.67 to 0.78 in asperity regions (Fig. 1c, d; Supplementary Fig. 1).

## Stress inversion

The applied stress inversion algorithm[59,60] is used to invert focal mechanisms for a set of earthquakes into tectonic stress, under the assumptions that a uniform regional stress field applies and that the earthquakes occur on faults with varying orientations, but they do not interact with each other. Given that there are only 2 $M_L > 5.0$ events, with the stronger earthquake being $M_L = 5.4$, the effects of co-seismic stress changes can be neglected[61,62] in our study. Under these assumptions, we invert focal mechanisms to estimate the orientation of the principal stress axes (and consequently, the principal focal mechanism) and the shape ratio, R, which determines the relative amplitude of the principal stresses. A typical problem in stress inversion from focal mechanisms concerns the selection of the actual fault plane from the two nodal planes, a choice which may severely bias the results[63] mainly the shape ratio determination. To avoid this effect, an algorithm which inverts jointly for stress and fault orientations is considered[64]. The fault orientation determination is based on the evaluation of fault instability index, I:

$$I = \frac{\tau + \mu(\sigma - 1)}{\mu + \sqrt{1 + \mu^2}} \quad (4)$$

where $\tau$ and $\sigma$ are the normalized shear and normal tractions, respectively and $\mu$, the friction coefficient. The fault plane is selected as the one having the highest $I$ value, after applying an iterative process. The iteration process performs a linearized inversion[59], each time introducing a different noise realization, which randomly rotate the given focal mechanisms. We applied 1000 such realizations in each case tested. The final stress tensor is derived as the mean of the results from all realizations. The uncertainty of the principal stress axes orientations is demonstrated by the scatter of the derived values (Supplementary Fig. 3a), whereas the shape ratio uncertainty is determined as the 95% confidence interval. The instability of principal stress axes shows the impact of the uncertainties from random perturbation of the solution, however it is not clear whether this instability arises exclusively from the spatial stress variations or uncertainties in the focal mechanism determination.

We use the available focal mechanisms along the Chain transform valley to determine the stress field. The vast majority of the reported GCMT solutions[49] suggest strike-slip faulting along Chain Transform Fault. However, our 1-year seismicity analysis of the OBS data reveals that a proportion of smaller magnitude events $(3.0 < M_L < 5.0)$ demonstrate a considerable vertical component, plausibly related with the positive flower structures (Fig. 1a). We derived 47 best constrained focal mechanisms that are considered for the stress inversion. The principal focal mechanism has strike = 275°, dip = 62° and rake = 165° (Supplementary Fig. 3b). We also derived a shape ratio, $R = 0.86$, with 95% confidence interval [0.69–0.89]. Dividing the region into smaller areas for a spatial analysis would significantly lower the significance of the results due to the decreasing sample size, therefore, we are not able to quantify the spatial variation of the stress along Chain.

The high $R$ value also suggests that the intermediate, $\sigma_2$ and minimum, $\sigma_3$ principal stresses, both deviate considerably from the vertical direction (plunge is 60° for $\sigma_2$ and 30° for $\sigma_3$) and they have similar amplitudes (Supplementary Fig. 3a, c). Therefore, they are virtually indistinguishable and might switch with each other. With $\sigma_1$ being clearly sub-horizontal (2° plunge), this physically means that under such stress field the simultaneous activation of both strike-slip and reverse faults is plausible[64]. This suggests a non-negligible reverse movement, related to the flower structures, especially within the ELFS, indicating the existence of active transpressional features along Chain.

## S-wave model from local Rayleigh and ambient noise tomography

We compare our result to the S-wave velocity structure derived from local Rayleigh wave and ambient noise tomography[35]. Group velocity measurements were inverted from 16 to 40 s period to determine 2-D velocity maps for each period of interest. Then the group velocities at each pixel across all of the maps were inverted for S-wave velocity structure. Additional details can be found in the original work. The lateral smoothing or correlation length in the group velocity map is 100 km, and checkerboard tests from the study suggest that anomalies at this length scale are well resolved by the method, particularly in the region around the Chain Transform Fault as it is within the center of the array, with many crossing ray paths with a wide range of azimuths. The formal resolution for the inversion for S-wave velocity with depth is 0.1 for 1 km thick layers in the upper 10 km beneath the seafloor and 20 km at greater depths. This suggests that the S-wave velocity inversion can uniquely resolve the average velocity over the upper 10 km of the model beneath the seafloor, the average velocity from 10 to 30 km beneath the seafloor and the average value from 30 to 50 km beneath the seafloor. Therefore, we do not interpret S-wave velocity model variations at any particular depth, rather in an average sense, over some depth range. The lateral and vertical resolution suggests that the variation in lithospheric thickness/velocity we observe across the transform fault at ~100 km scale (WFS1-WFS3, A2 vs ELFS) and over 50 km depth is robust. The broader thermal structure of the lithosphere in the region is well within the resolving power of the model presented here. The low P-wave velocities just beneath the Moho of the ELFS from the active source model do not appear in the S-wave velocity model derived from the surface waves, although this is expected given the different resolution of the waveforms. The scale of the crustal and shallow mantle variations observed in our P-wave refraction model is ~25–30 km laterally along strike of the transform, 10–20 km across the transform and occurring over ~5–10 km in depth. This is below the resolution of the tomographic model used here as described above. In other words, the velocity anomalies in the refraction model would be averaged out in the S-wave velocity model with the relatively faster lithosphere on either side of the fault zone.

The 600 °C isotherm derived from the S-wave velocity derived model is closer to 30 km depth across the entire transform, deeper and different in shape than that of simple thermal models (Fig. 3). The scale of this variation, 100 km laterally, is well resolved as described above. Isotherms with shapes that deviate from those of simple thermal models can be caused by upwelling in the center of the transform, as predicted by geodynamic modeling with a visco-elastic-plastic rheology[6]. While such a rheology cannot be precluded here, we also require an additional factor, given that our isotherms are also deeper than predicted by those models.

## S-wave model translated into thermal model

We examine two methods for translating seismic S-wave velocity to temperature based on the assumption of a peridotitic mantle. In the main text we use the empirical fit to predictions for a pyrolite mantle[47]:

$$V_s = 4.77 + 0.038P - 0.00038(T - 300) \quad (5)$$

where $V_s$ is S-wave velocity in km/s, $P$ is pressure in GPa and $T$ is temperature in Kelvin. We also investigate the effects of different composition assumptions on the conversion from S-wave velocity to temperature. We use the Abers and Hacker[65] MATLAB toolbox to calculate the predicted S-wave velocity for three end member compositions. We use a depleted upper mantle (harzburgite), an undepleted mantle (lherzolite) and, for reference, an idealized unmelted mantle composition (pyrolite). The specific mineral modes we use are as follows: harzburgite (fo 72.48% volume, fa 7.52%, en 18.24%, fs 1.76%), lherzolite (fo 45.41%, fa 5.07%, en 22.48%, fs 2.50% di

20.03, sp 4.50%) and pyrolite (alm 2.10%, gr 1.10%, py 10.80%, fo 54.50%, fa 6.70%, en 14.70%, fs 1.50%, di 7.50%, hed 1.2%). We do not account for phase changes of the aluminious phases in the calculations although the spinel – garnet transition is encompassed somewhat between the lherzolite and pyrolite models. We apply a frequency dependent attenuation correction to the S-wave velocity output from the codes using both the approach and the 1-D attenuation structure[66]. The velocities were calculated for a range of temperatures at a given pressure corresponding to the depth in the model and the temperature corresponding to each velocity in the model was determined by interpolation.

The overall effect of choosing a different starting composition shallows the isotherms predicted from the S-wave velocity model relative to the Stixrude and Lithgow-Bertelloni[47] parameterization, particularly the 600 °C isotherms, although the effect is not substantial. With the Abers and Hacker[65] parameterization the 700 °C isotherm lies at nearly the same depth as the 600 °C isotherm from the Stixrude and Lithgow-Bertelloni[47] parameterization. The difference between the compositions from the Abers and Hacker[65] calculations is small, typically the isotherms are within a few km of each other. If the compositions and calculations from Hacker are used all the well-located events are shallower than the 700 °C isotherm.

### Seismic tomography of active source airgun profiling

We used data collected at the easternmost section of the Chain Fracture Zone as part of the ILAB-SPARC experiment[67] (Supplementary Data 1; Supplementary Figs. 4–6). Seismic data were acquired in October of 2018 aboard the French research vessel Pourquoi Pas?. A 140 km long seismic profile was acquired at the eastern Ridge-Transform Intersection running from the pressure ridge into the adjacent fracture zone (Supplementary Fig. 4). Five OBS spaced every 12 km recorded airguns shots fired with the Pourquoi Pas?; two sub-arrays containing eight G-guns each provided a total volume of 82 litres and were towed at a depth of 10 m.

Travel times of first arrival P-waves and secondary wide-angle-refection interpreted to be crust/mantle boundary reflection (PmP) arrivals are hand-picked. In general, picking uncertainties were 20–30 ms for short-offset P-waves (Pg) and reach 60 ms for far-offset P-waves (Pn) and secondary PmP arrivals.

Seismic refraction travel time data were used to derive 2-D velocity models using a seismic tomography approach[68]. The method employs a hybrid ray-tracing scheme combining the graph method with further refinements utilizing ray bending with the conjugate gradients method. Smoothing and damping constraints regularize the iterative inversion. Details of the procedure are given elsewhere[40]. Picking errors and starting velocity models may control inversion results. We therefore chose a nonlinear Monte Carlo-type error analysis to derive model uncertainties. The approach consists of randomly perturbing the velocity values of an initial average 1-D model to create a set of 100 2-D reference models, providing a well constrained seismic velocity model (Supplementary Fig. 5).

Our P-wave refraction study illuminates anomalous crust and mantle structure beneath ELFS. There is a clear Moho reflector visible across the region, which is relatively flat at 10 km depth, especially beneath ELFS (Supplementary Fig. 6, Fig. 4a). Within the crust of ELFS there is evidence of a high velocity core, while the crustal structure away from ELFS has relatively flat velocity contours (Fig. 4a). In the upper most mantle beneath the Moho reflector we observe a broad region of slower than expected mantle velocities (<7.5 km/s), i.e., 4–10% slower in comparison to velocities >7.8 km/s on the western side of the ELFS. The slow velocity anomaly extends to at least 15 km depth and is centered just east of ELFS.

## Data availability

The earthquake catalog data used in the study are available in Schlaphorst et al.[33]. The ocean bottom seismic data used in the study are archived and can be obtained at the IRIS DMC, as 2016–2017 network XS https://doi.org/10.7914/SN/XS_2016[69]. The relocated data since 1993[31] used in the study are available at https://doi.org/10.5281/zenodo.4646438. The bathymetry and gravity models are available at [ftp://ftp.noc.soton.ac.uk/pub/nh1v08.]. The active source velocity model generated in this study is provided as Supplementary Data 1 file.

## Code availability

Some figures were generated using Generic Mapping Tools v.6.1.1 (www.soest.hawaii.edu/gmt, last accessed March 2023). The MATLAB code for the Anderson-Darling test can be downloaded from https://git.plgrid.pl/projects/EA/repos/sera-applications/browse/. The STRESSINVERSE package is available in MATLAB and Python at https://www.ig.cas.cz/stress-inverse/. The MATLAB toolbox for calculating the predicted S-wave velocities is available at https://agupubs.onlinelibrary.wiley.com/doi/10.1002/2015GC006171.

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

## Acknowledgements

The authors acknowledge funding from the Natural Environment Research Council (NE/M003507/1) (K.L., C.A.R and N.H.) (NE/M004643/1) (J.M.K), the European Research Council (GA 638665) (K.L., C.A.R. and N.H.) and the National Science Foundation (grants NSF-EAR-2147918 and NSF-OCE-231613) (C.A.R). We thank the captain and crew of the R/V Marcus G. Langseth and the RRS Discovery, and the scientific technicians. The active source seismic data were acquired during the ILPAB-SPARC experiment on board the French R/V Pourquoi Pas? funded by the European Research Council Advanced Grant number 339442_TransAtlanticILAB. Refraction data were analyzed by Christian Filbrandt. We thank the editor, Dr. Sebastian Mueller and 4 anonymous reviewers for the constructive comments and suggestions.

## Author contributions

K.L., C.A.R. and N.H. developed the main ideas, performed analyses (magnitude distribution, statistical analyses, seismic coupling, velocity/thermal models, stress inversion) and generated figures. K.L. produced the initial manuscript. D.S. analyzed the OBS data to determine locations, magnitudes and focal mechanisms. I.G. provided the P-wave refraction model for the eastern part of Chain. K.L., C.A.R., N.H., D.S., I.G., J.-M.K. and S.C.S. discussed the results and contributed to writing the manuscript.

## Competing interests

The authors declare no competing interests.
