## [Peer Review File · Nature Communications]

Broad fault zones enable deep fluid transport and limit earthquake magnitudesREVIEWER COMMENTS

Reviewer #1 (Remarks to the Author):

This paper describes a study that investigates the characteristics of seismicity that occurs in the Chain Fracture Zone, an oceanic transform fault. In particular, it looks at how the magnitude distribution changes spatially with characteristics such as morphological features identified in bathymetry and gravity data. It finds that the Chain can be divided into 2 asperity and 3 barrier zones. The barrier zones are characterized by smaller moment release than expected and higher b-values (i.e., relatively more smaller earthquakes than larger ones). They also tend to occur where the flower structures and therefore broader fault zones are located, which suggests that these features facilitate fluid transport deeper into the lithosphere and weakening it, thus reducing the potential for large earthquakes to occur.

I very much enjoyed reading this paper; I thought it was well-written and well-reasoned, and easy to follow. I appreciated the thoroughness with which the b-values were investigated, especially considering multiple methods. It is certainly on appropriate topics for this journal, combining earthquake statistics with seafloor geophysics observations and modeling, and I think a broad multidisciplinary range of readers will appreciate it. What controls earthquake rupture size is of course a topic of great interest with implications for seismic hazard, and this study could be a valuable contribution to that topic, by examining it in the relatively simpler oceanic transform fault environment. I have just a few minor comments and suggestions listed below.

p. 1, 2nd paragraph: What magnitude would the rupture of the entire transform fault length potentially result in? (to compare it numerically with M7)

p. 3, Fig. 1 caption: The caption for panel (b) states that black dots are in low rMBA and white dots are in high rMBA, but the figure itself shows the reverse. I think the caption might have just accidentally switched them.

p 4, Fig. 2: I think it could help to label the isotherms that are shown in panels (b) and (c). I'm not exactly sure which one is the 600C isotherm, but I suppose I could guess from the location of the red dots. Also, the caption states that "the events located deeper than the 600C isotherm from simple thermal models are highlighted with a red dot." Is this the thermal model shown in panels (b) and (c)? Finally, the caption for (b) states that "the dashed red isolines indicate the thermal structure..." but I think these are actually shown in blue, correct?

p. 11, section on OBS seismicity (and Fig. 3d and e): The red and blue distributions in these panels do look qualitatively different. Have you done any statistical testing to show that they are quantitatively significantly different?

Extended Data Fig. 7: It might be informative to indicate in this figure where the nominal M_c that you use in this study lies ($M_c=2.3$), perhaps with a vertical dashed line.

p. 12: I believe that the numerator for the equation for b_{ij} should actually be " $\log_{10}(N_j)-\log_{10}(N_i)$ "

Reviewer #2 (Remarks to the Author):

The authors present a seismological study of the Chain transform fault in the mid-Atlantic ocean ridge. Combining several different datasets together, the authors argue that morphological features of the fault zone, primarily flower structures linked to step overs in the fault system, facilitate the circulation of fluids into the fault zone, pushing the brittle-ductile transition zone deeper and acting as barriers to major seismic ruptures. The manuscript is well structured and easy to read (commendable given the amount of different datasets involved!) and the figures informative. While the interpretation is not necessarily that surprising, the authors' argument is well justified from the unique combination of data types exploited here. The manuscript is thus of sure interest to the community.

The technical approach and analysis seem sound to me, as the methods are for the most part standard techniques. One main comment would be on the proposed interpretation of the data that fails to address the question of fault heterogeneity. The prominence of microseismicity at the apparent barriers to seismic slip would suggest to me that the distribution of heterogeneity length scales is significantly different between "asperities" (where large earthquakes have occurred) and "barriers". The text makes reference to "mechanical and chemical effects...increasing pore pressure" (L183) caused by the presence of fluids in the fault zone, but this vague explanation is likely only part of the story. A reduction of effective normal stress would indeed favor seismic rupture, but how the rupture plays out after nucleation is controlled by heterogeneity (whether structural or in the stress state). Given that we know that fluids (water) can alter lithologies, it would be logical to me that it would also affect the distribution of fault heterogeneity, potentially leading to favoring a distributed pattern of seismic release (microseismicity) rather than large fault-wide ruptures.

Beyond this point of discussion that I think should be addressed, I have a few other minor comments to make.

L23 - punctuated locations (barriers) -> localized barriers

L52 - isn't microseismicity brittle rupture?

Fig 1 - vertical alignment of axes labels is quite strange; should likely just rotate 90 degrees as is common

Fig 1 - It would be quite helpful to show a panel with cumulative seismic moment as a function of along-strike position. This would clearly highlight where potential asperities and barriers are. This would also complement well the seismic moment percentages

L131 - red -> blue (?)

Fig 3a - why are there cyclic variations in the b-values as a function of bathymetric depth?

Fig 3b and c - Why not use the same color scheme for these two panels? It would make it clearer that the annotations in b are the fits to the distributions in c

Reviewer #3 (Remarks to the Author):

This article outlines a series of observations that the authors propose support an evolving theory that hydrothermal processes lead to earthquake segmentation of oceanic transform faults. Much of the reasoning in the article leans heavily on recent work done at other fault systems, particularly that there are distinct rupture patches on transform faults adjacent to rupture barriers, and that this leads to the overall low seismic coupling seen at oceanic transform faults. New observations from the Chain transform are presented, which is a valuable comparison to add to a growing number of regional studies using marine seismic and OBS data.

The authors suggest that the barrier regions can be associated on the Chain transform fault with transpressional flower structures, interpreted in (undefined resolution) multibeam bathymetry data. They also reference evidence for hydrothermal processes in these zones from a few observations including 1. Several (8?) locatable deep earthquakes that occur within the barrier zones, 2. low RMBA gravity anomalies within the barrier zones, possibly associated with alteration?, and 3. seismic velocity models and estimated thermal structures from these models. Although I appreciate the new set of observations that the authors present in this paper, and I believe much of the logic they present is good for arguing for the link between earthquake rupture properties and fluid-rock interactions, I do not feel that the data set shows the correlations they outline clearly as it is presented. My primary concerns stem from a few specific things: 1. I am concerned that the multibeam bathymetry images are over interpreted, and that 'flower structures' identified by the authors may be coarse-resolution expressions of a more complex structural configuration that is accommodating variation in hydrogeology or some other source of material heterogeneity, 2. Inferences made regarding along-strike changes in thermal structure based on seismic velocities seem to be strangely presented and interpreted. Seismic velocity models shown in the paper do not seem to show striking low velocity zones where the authors claim there are altered zones. (Or I may just be having a difficult time making the needed correlation b/w seismic velocity and rupture zones) 3. Geophysical trends in gravity from RMBA are presented as evidence

of hydrothermal processes, but it's not clear to me that they necessarily support this interpretation. Forward modeling of how low RMBA could relate to (for example) serpentinization or porosity (not mentioned) as well as thickened crust (how does this promote hydrothermal cooling?) would be helpful to strengthen this interpretation and clarify how low RMBA could link to material variability that results from the proposed hydrothermal processes.

Below I have outlined some specific sections of the manuscript that I found to be confusing or in need of reconsideration:

- Line 70: "The Chain TF contains four lozenge-shaped topographic highs and shows evidence for variability in gravity anomalies and inferred crustal structure along the length of the TF8 (Fig 1a, b). These features represent crustal blocks exhumed via transpressional stresses, i.e., features typically referred to as positive flower structures (Methods). "

I am not certain, as the it is currently presented that the presence of these flower structure is very compelling. Figure 1a shows bathymetry (displayed at an unlabeled ? resolution) with the topographic highs interpreted. My skepticism comes from the seeming likelihood that these features are being over-interpreted based on the apparently low-resolution dataset and the horizontal earthquake uncertainties as well. What is the resolution of the seafloor morphology that is being presented in the bathymetry? Are we sure that topographical highs and earthquake patters are really correlating with the uncertainties of both datasets? Could there be finer scale features (i.e. fault step-overs, multiple strands, as the authors point out, but also extensional bends or wide damage zones?) that are equally responsible for the observed heterogeneity in earthquake behavior along strike? It's not clear how robust the morphology interpretations are, and it seems like they do not perfectly correlate with either the RMBA or the earthquake locations of more-frequent smaller events.

- Line 192: "The S-wave velocity model predicts that the 600°C isotherm associated with the brittle-ductile transition occurs deeper than predicted for a thermal model, supporting the hypothesis that deeper cooling has occurred. "

It seems like the authors may need to spend more time convincing us that the interpretation of S-wave velocity is a function of temperature, and not other physical properties. The S-wave velocity structure does not predict that it is cooler in the barrier zone relative to the asperitiy zone, just that the temperatures are cooler than predicted by a (very simple!) half space cooling model. More advanced numerical models predict a cooler thermal structure both near the transform fault ends (flatter isotherms) and if hydrothermal cooling is present... this is not a surprise anywhere along a transform fault, but doesn't seem to correlate with earthquake barrier zones shown here, except possibly below 30 km, and well below the 900° C isotherm based on S-wave velocities (where we would not expect brittle failure?)

- Line 201: "In addition, the predicted temperatures from the S- wave velocity model are hotter at 25 – 30 km depth than surrounding regions, "

Similar to comment above: This doesn't seem to be true based on either Figure 2b or 2c? I don't see how the S-wave velocity model shows an anomaly in this range, or that there is a significant thermal anomaly (predicted from the S-wave velocity model) that is consistent with this. I wouldn't expect that S-wave anomalies would necessarily be caused by changes in temperature as opposed to lithologic variability or variation in porosity.

- Line 177: "Hydrothermal alteration of the shallow lithospheric mantle (<15 km beneath the sea surface) may be suggested, given the anomalously slow sub-crustal P-wave velocities down to at least 15 km"

I am finding it difficult to interpret Figure 2a showing the P-wave velocity. Even as someone who models crustal velocity structure, it seems like this P-wave model is only a little bit slower than typical crust at ~8 km below the surface (sub-crustal depths?) – with no reference, it's difficult to be sure of what the velocities mean in here. Also where is this model exactly relative to the Chain TF? Does it extend past the ridge – to the east of the fault shown in b and c panels? Although I think it is useful to have a velocity model if the authors are arguing for the impact of hydrothermal alteration, in this form, I am not sure I gain much in the way of evidence for hydrothermal processes from this figure.

- Line 180: "Alteration caused by fluid infiltration may also explain why the section containing the longest (50 km) fault scarp in the region is not associated with the largest earthquakes, but instead associated with elevated b-values (1.1 – 1.2) with very low amounts of energy released seismically (4 %)."

This is quite confusing. Are the authors saying here that hydrothermal alteration is prevalent everywhere along the fault – in the asperity and barrier zones? Where specifically are they referring to when they reference the 50 km long fault scarp (is this within the AX or BX zones?).

- Re Methods - S-wave model translated into thermal model: I am not an expert in using S-wave velocities to estimate thermal structure, but it does seem to me that there could be physical factors other than just temperature that are influencing S-wave velocity at relatively shallow depths, such as the 5-20 km below the seafloor depths that are shown in the paper overlapping with seismicity.

REVIEWER COMMENTS

Reviewer #1 (Remarks to the Author):

This paper describes a study that investigates the characteristics of seismicity that occurs in the Chain Fracture Zone, an oceanic transform fault. In particular, it looks at how the magnitude distribution changes spatially with characteristics such as morphological features identified in bathymetry and gravity data. It finds that the Chain can be divided into 2 asperity and 3 barrier zones. The barrier zones are characterized by smaller moment release than expected and higher b-values (i.e., relatively more smaller earthquakes than larger ones). They also tend to occur where the flower structures and therefore broader fault zones are located, which suggests that these features facilitate fluid transport deeper into the lithosphere and weakening it, thus reducing the potential for large earthquakes to occur. I very much enjoyed reading this paper; I thought it was well-written and well-reasoned, and easy to follow. I appreciated the thoroughness with which the b-values were investigated, especially considering multiple methods. It is certainly on appropriate topics for this journal, combining earthquake statistics with seafloor geophysics observations and modeling, and I think a broad multidisciplinary range of readers will appreciate it. What controls earthquake rupture size is of course a topic of great interest with implications for seismic hazard, and this study could be a valuable contribution to that topic, by examining it in the relatively simpler oceanic transform fault environment. I have just a few minor comments and suggestions listed below.

p. 1, 2nd paragraph: What magnitude would the rupture of the entire transform fault length potentially result in? (to compare it numerically with M7)

We added the following sentence:

“For instance, at the Chain TF the magnitude of rupturing the entire transform is predicted to be $M > 8.0^{13}$ based on empirical relationships ”

With ¹³: Wells and Coppersmith, 1994

p. 3, Fig. 1 caption: The caption for panel (b) states that black dots are in low rMBA and white dots are in high rMBA, but the figure itself shows the reverse. I think the caption might have just accidentally switched them.

Yes, fixed.

p 4, Fig. 2: I think it could help to label the isotherms that are shown in panels (b) and (c). I'm not exactly sure which one is the 600C isotherm, but I suppose I could guess from the location of the red dots.

We increased the size of the contour labels and framed them within blue boxes to make them more visible.

Also, the caption states that “the events located deeper than the 600C isotherm from simple thermal models are highlighted with a red dot.” Is this the thermal model shown in panels (b) and (c)?

We added the parenthetical reference in the following sentence in the figure caption:

“The events located deeper than the 600°C isotherm from simple thermal models (dashed blue lines in panels b and c) are highlighted with a red dot.”

Finally, the caption for (b) states that “the dashed red isolines indicate the thermal structure...” but I think these are actually shown in blue, correct?

We changed 'red dashed' to 'blue dashed.'

p. 11, section on OBS seismicity (and Fig. 3d and e): The red and blue distributions in these panels do look qualitatively different. Have you done any statistical testing to show that they are quantitatively significantly different?

We added the following to the text of the methods section:

“We perform two statistical tests, to quantify the difference between the two distributions. The Wilcoxon rank sum test is used to test the null hypothesis that the data in the two sets (i.e., epicentres and background data) are drawn from distributions with equal medians. In addition, the two sample Kolmogorov-Smirnov test was applied to test the hypothesis that the data in the two sets (i.e., epicentres and background data) are drawn from the same

continuous distributions. Both Wilcoxon rank sum test and Kolmogorov-Smirnov test reject the corresponding null hypotheses: at $p < 10^{-6}$ for rMBA and at $p < 10^{-4}$ for bathymetry.”

Extended Data Fig. 7: It might be informative to indicate in this figure where the nominal M_c that you use in this study lies ($M_c=2.3$), perhaps with a vertical dashed line.

Done (now Supplementary material Fig. 8).

p. 12: I believe that the numerator for the equation for b_{ij} should actually be “ $\log_{10}(N_j) - \log_{10}(N_i)$ ”

Corrected.

Reviewer #2 (Remarks to the Author):

The authors present a seismological study of the Chain transform fault in the mid-Atlantic ocean ridge. Combining several different datasets together, the authors argue that morphological features of the fault zone, primarily flower structures linked to step overs in the fault system, facilitate the circulation of fluids into the fault zone, pushing the brittle-ductile transition zone deeper and acting as barriers to major seismic ruptures. The manuscript is well structured and easy to read (commendable given the amount of different datasets involved!) and the figures informative. While the interpretation is not necessarily that surprising, the authors' argument is well justified from the unique combination of data types exploited here. The manuscript is thus of sure interest to the community.

The technical approach and analysis seem sound to me, as the methods are for the most part standard techniques. One main comment would be on the proposed interpretation of the data that fails to address the question of fault heterogeneity. The prominence of microseismicity at the apparent barriers to seismic slip would suggest to me that the distribution of heterogeneity length scales is significantly different between "asperities" (where large earthquakes have occurred) and "barriers". The text makes reference to "mechanical and chemical effects...increasing pore pressure" (L183) caused by the presence of fluids in the fault zone, but this vague explanation is likely only part of the story. A reduction of effective normal stress would indeed favor seismic rupture, but how the rupture plays out after nucleation is controlled by heterogeneity (whether structural or in the stress state). Given that we know that fluids (water) can alter lithologies, it would be logical to me that it would also affect the distribution of fault heterogeneity, potentially leading to favoring a distributed pattern of seismic release (microseismicity) rather than large fault-wide ruptures.

We revised the final paragraph to add more details: "The Chain TF provides a view into how segmentation on oceanic TF can develop, which may in turn explain why oceanic TF do not generate larger earthquakes for their fault length. Multiple overlapping faults located on the flower structures are associated with broader damage zones, which provide a wider pathway for fluids into the crust and uppermost mantle. The presence of fluids causes a series of mechanical and chemical effects, potentially increasing the pore pressure, reducing the effective normal stresses, lubricating the fault, and/or creating weaker regions of altered material. The hydrothermal circulation that occurs at shallow depths also cools the underlying mantle deepening the brittle ductile transition. However, the damaged/altered crust and upper mantle are weaker, permitting only low magnitude earthquakes and/or aseismic accommodation of tectonic strain accumulation in these barrier regions beneath the flower structures. The barrier regions separate stronger, locked regions (asperities) beneath singular, linear fault strands with narrower damage zones, where hydrothermal circulation is lower, the brittle ductile transition and observed earthquake depths are shallower. The faults in the asperity regions are stronger, permitting larger events. In other words, broader fault zones can effectively segment the fault."

Beyond this point of discussion that I think should be addressed, I have a few other minor comments to make.

L23 - punctuated locations (barriers) -> localized barriers

Fixed

L52 - isn't microseismicity brittle rupture?

Changed to 'rather than large scale ruptures'

Fig 1 - vertical alignment of axes labels is quite strange; should likely just rotate 90 degrees as is common

We added the following explanation to the figure caption: We use an oblique Mercator projection so that the transform appears horizontal, which is the reason for angled ticks marking longitude and latitude.

Fig 1 - It would be quite helpful to show a panel with cumulative seismic moment as a function of along-strike position. This would clearly highlight where potential asperities and barriers are. This would also complement well the seismic moment percentages

Done. We added an extra panel (d) in figure 1, showing the along-strike cumulative seismic moment of the historical events.

L131 - red -> blue (?)

Fixed

Fig 3a - why are there cyclic variations in the b-values as a function of bathymetric depth?

The b-value drops are caused by the stronger events, which makes sense since the windows have only 100 events, making the calculations sensitive to the existence of strong events. What really matters is the general decreasing trend of the b-value. We added the following to the text:

“The apparent cyclic relationship between the b-values and bathymetry is an artefact related to the low number of events in each calculation (100). b-values calculated using a small number of data are more sensitive to the larger magnitude events, which coincide with the x-axis locations where jumps in b-value are apparent (Fig. 3a).”

Fig 3b and c - Why not use the same color scheme for these two panels? It would make it clearer that the annotations in b are the fits to the distributions in c

Fixed

Reviewer #3 (Remarks to the Author):

This article outlines a series of observations that the authors propose support an evolving theory that hydrothermal processes lead to earthquake segmentation of oceanic transform faults. Much of the reasoning in the article leans heavily on recent work done at other fault systems, particularly that there are distinct rupture patches on transform faults adjacent to rupture barriers, and that this leads to the overall low seismic coupling seen at oceanic transform faults. New observations from the Chain transform are presented, which is a valuable comparison to add to a growing number of regional studies using marine seismic and OBS data.

We agree that our observations are complimentary to some previous work. However, our reasoning and conclusions do not rely on it, and we have altered the text to better clarify this. Much previous work relies on 1 or 2 observational constraints, e.g., seismicity or seismicity and gravity, etc. We are showing new independent velocity models that provide independent support for hydrothermal circulation and cooling beneath regions with wide fault deformation zone. This kind of association and process are not part of previous work that we are aware of. Our work also provides a possible explanation for the cause of the barrier regions, in other words that broader fault damage zones preferentially facilitate fluid transport into the lithosphere.

We reworded the following paragraph to increase clarity:

“A few other studies have reported TF seismicity at deeper depths than the 600°C isotherm¹⁴⁻¹⁶. Geodynamic models predict a deepened 600°C isotherm if there is hydrothermal alteration, and this is true particularly near the ends of the transform²². However, seismicity has also been reported at deeper depths than predictions including hydrothermal alteration^{15,22}. Therefore, it has been suggested that this can be explained by higher strain rates at transform faults³⁸. Alternatively, different rheological flow laws may be in effect, and brittle and ductile deformation may occur over a broad range of temperatures (300°C – 1000°C) and, therefore, a broad range of depths, owing to variable seawater infiltration and grain sizes along the fault²³. Our result does not preclude the possibility that either or both of these two scenarios are in effect in some locations globally. However, what is different about our work is that our S-wave velocity model offers an independent constraint that is not often available. It shows that different flow laws and/or high strain rates are not required at Chain, since seismicity occurs shallower than the 600°C isotherm derived from the S-wave velocity model. In other words, our S-wave velocity model demonstrates that hydrothermal circulation has cooled our region more efficiently and to deeper depths beneath the flower structures with multiple fault strands and enhanced damage and the likelihood of large earthquakes, and this model is likely in effect elsewhere.”

We also added the following sentence:

“Although previous studies have suggested that transforms may be segmented, and it has been hypothesized that variable water infiltration into the lithosphere plays a role, our study provides new constraints on the Earth properties that can result in such phenomena.”

The authors suggest that the barrier regions can be associated on the Chain transform fault with transpressional flower structures, interpreted in (undefined resolution) multibeam bathymetry data. They also reference evidence for hydrothermal processes in these zones from a few observations including 1. Several (8?) locatable deep earthquakes that occur within the barrier zones, 2. low RMBA gravity anomalies within the barrier zones, possibly associated with alteration?, and 3. seismic velocity models and estimated thermal structures from these models. Although I appreciate the new set of observations that the authors present in this paper, and I believe much of the logic they present is good for arguing for the link between earthquake rupture properties and fluid-rock interactions, I do not feel that the data set shows the correlations they outline clearly as it is presented. My primary concerns stem from a few specific things: 1. I am concerned that the multibeam bathymetry images are over interpreted, and that ‘flower structures’ identified by the authors may be course-resolution expressions of a more complex structural configuration that is accommodating variation in hydrogeology or some other source of material heterogeneity, 2. Inferences made regarding along-strike changes in thermal structure based on seismic velocities seem to be strangely presented and interpreted. Seismic velocity models shown in the paper do not seem to show striking low velocity zones where the authors claim there are altered zones. (Or I may just be having a difficult time making the needed correlation b/w seismic velocity and rupture zones) 3. Geophysical trends in gravity from RMBA are presented as evidence of hydrothermal processes, but it’s not clear to me that they necessarily support this interpretation. Forward modeling of how low RMBA could relate to (for example) serpentinization or porosity (not mentioned) as well as thickened crust (how does this promote hydrothermal cooling?) would be helpful to strengthen this interpretation and clarify how low RMBA could link to material variability that results from the proposed hydrothermal processes.

We thank the reviewer for this useful query. We have responded to the content of this paragraph in the more detailed review sections below.

Below I have outlined some specific sections of the manuscript that I found to be confusing or in need of reconsideration:

- Line 70: “The Chain TF contains four lozenge-shaped topographic highs and shows evidence for variability in gravity anomalies and inferred crustal structure along the length of the TF8 (Fig 1a, b). These features represent crustal blocks exhumed via transpressional stresses, i.e., features typically referred to as positive flower structures (Methods). “
I am not certain, as the it is currently presented that the presence of these flower structure is very compelling. Figure 1a shows bathymetry (displayed at an unlabeled ? resolution) with the topographic highs interpreted. My skepticism comes from the seeming likelihood that these features are being over-interpreted based on the apparently low-resolution dataset and the horizontal earthquake uncertainties as well. What is the resolution of the seafloor morphology that is being presented in the bathymetry? Are we sure that topographical highs and earthquake patters are really correlating with the uncertainties of both datasets?

We added the horizontal resolution of the bathymetry (100 m) and vertical resolution of the bathymetry (~10s of m) of meters to the figure caption. These uncertainties are an order of magnitude smaller than the size of the structures interpreted in the bathymetry. For instance, the width of the transform is ~15 km.

The argument for transpression is presented in the Supplementary material section. We added some more information, commenting on the earthquake focal mechanisms (*italics*, below) in more detail:

“**Origin of the lozenge-shaped topographic highs.** One possibility is that these features represent intra-transform volcanism³⁹. However, the observed low backscatter intensity of these structures⁸ does not support recent volcanism. A more likely scenario is that these features represent crustal blocks exhumed via transpressional stresses underlain by altered mantle, i.e., features typically referred to as positive flower structures. *This is also supported by the 11 reverse fault mechanisms recorded by the OBS along the transform (Fig. 1). There were only 2 events characterized by a dominantly normal focal mechanism, potentially explained by flank collapses.* Similar morphological features are frequently observed and fault plane locations are sometimes available to substantiate a transpressional origin. This is supported by the highly tectonised mafic and ultra-mafic compositions of rocks from features with similar morphology and gravity signatures at the nearby St. Paul TF that are also thought to have a transpressional origin⁴⁰.”

Could there be finer scale features (i.e. fault step-overs, multiple strands, as the authors point out, but also extensional bends or wide damage zones?) that are equally responsible for the observed heterogeneity in earthquake behavior along strike?

Yes, we agree that there are multiple fault strands visible in the bathymetry (Supplementary Material Fig. 1). They support our hypothesis that the flower structures are wide damage zones that contribute to the heterogeneity.

We added the following to the text, “There could be additional fault structures beneath the resolution of our bathymetry, but there is no reason to believe that they would be different or more important than the large-scale structures on which we have based our arguments.”

We added the following sentence, “The length scale of heterogeneity in seismicity is 10s of kms, which is much greater than the scale of bathymetric resolution (100 m), allowing us to make meaningful comparisons.”

We are interpreting the features at the scale that we can resolve. We see no evidence for transtension in the focal mechanisms of the earthquakes along strike.

It’s not clear how robust the morphology interpretations are, and it seems like they do not perfectly correlate with either the rMBA or the earthquake locations of more-frequent smaller events.

Please see other responses that describe the resolution of the bathymetry.

- Line 192: “The S-wave velocity model predicts that the 600°C isotherm associated with the brittle-

ductile transition occurs deeper than predicted for a thermal model, supporting the hypothesis that deeper cooling has occurred. “
It seems like the authors may need to spend more time convincing us that the interpretation of S-wave velocity is a function of temperature, and not other physical properties.

We added the following to the text:

“Seismic velocities can be influenced by a variety of factors such as composition including degree of alteration/hydration, pore fluid, or temperature. Typically, the bulk composition of depleted mantle is expected to be relatively homogeneous, but even large variations in depletion would yield very minor amounts of seismic velocity anomalies³³. Hydrothermal alteration and serpentinization are expected to reduce seismic velocities. However, the S-velocities we observe in the lithosphere are relatively high, 4.55 km/s on average, and inconsistent with these factors³⁴. Also, porosity effects on S-wave speeds are not likely to be important at the mantle depths where surface waves have sensitivity³⁵. Therefore, we proceed interpreting the S-wave velocities in terms of temperature. Observed slow P-wave velocities could be explained by thickened crust, enhanced porosity, and/or hydrothermal alteration, which we will discuss in greater detail in the next section.”

The S-wave velocity structure does not predict that it is cooler in the barrier zone relative to the asperity zone, just that the temperatures are cooler than predicted by a (very simple!) half space cooling model. More advanced numerical models predict a cooler thermal structure both near the transform fault ends (flatter isotherms) and if hydrothermal cooling is present... this is not a surprise anywhere along a transform fault, but doesn't seem to correlate with earthquake barrier zones shown here, except possibly below 30 km, and well below the 900° C isotherm based on S-wave velocities (where we would not expect brittle failure?)

We revised the following paragraph to improve clarity:

“A few other studies have reported TF seismicity at deeper depths than the 600°C isotherm¹⁴⁻¹⁶. Geodynamic models predict a deepened 600°C isotherm if there is hydrothermal alteration, and this is true particularly near the ends of the transform²². However, seismicity has also been reported at deeper depths than predictions including hydrothermal alteration^{15,22}. Therefore, it has been suggested that this can be explained by higher strain rates at transform faults³⁸. Alternatively, different rheological flow laws may be in effect, and brittle and ductile deformation may occur over a broad range of temperatures (300°C – 1000°C) and, therefore, a broad range of depths, owing to variable seawater infiltration and grain sizes along the fault²³. Our result does not preclude the possibility that either or both of these two scenarios are in effect in some locations globally. However, what is different about our work is that our S-wave velocity model offers an independent constraint that is not often available. It shows that different flow laws and/or high strain rates are not required at Chain, since seismicity occurs shallower than the 600°C isotherm derived from the S-wave velocity model. In other words, our S-wave wave velocity model demonstrates that hydrothermal circulation has cooled our region more efficiently and to deeper depths beneath the flower structures with multiple fault strands and enhanced damage, and this model is model is likely in effect elsewhere.”

We added the italicized sentence to the following paragraph in the methods section:

“The formal resolution for the inversion for S-wave velocity with depth is 0.1 for 1 km thick layers in the upper 10 km beneath the seafloor and 20 km at greater depths. This suggests that the S-wave velocity inversion can uniquely resolve the average velocity over the upper 10 km of the model beneath the seafloor, the average velocity from 10-30 km beneath the seafloor and the average value from 30-50 km beneath the seafloor. *Therefore, we do not interpret shear velocity model variations at any particular depth, rather in an average sense, over some depth range.*”

Given surface wave sensitivity it doesn't matter whether the 600 or 900 isotherm dips lower. We added the following in the discussion:

“Indeed, surface wave-derived shear velocities indicate higher velocities on average over the shallow mantle (down to 40 km depth) beneath the flower structures (Fig. 2). The predicted thermal contour depths (e.g., 600 °C) and shapes (e.g., 900 °C) are much different than the predictions for a simple conductive cooling thermal model, accounting for the depth sensitivity of surface wave inversion (Methods).”

- Line 201: “In addition, the predicted temperatures from the S- wave velocity model are hotter at 25 – 30 km depth than surrounding regions, “

Similar to comment above: This doesn't seem to be true based on either Figure 2b or 2c? I don't see how the S-wave velocity model shows an anomaly in this range, or that there is a significant thermal anomaly (predicted from the S-wave velocity model) that is consistent with this. I wouldn't expect that S-wave anomalies would necessarily be caused by changes in temperature as opposed to lithologic variability or variation in porosity.

This is similar the previous response. We also added the italicized phrase below to clarify:

"In addition, the predicted temperatures from the S-wave velocity model averaged over depth, given the sensitivity of surface waves (Methods), are hotter than surrounding regions which could be consistent with less hydrothermal cooling in this region (Fig. 2b)."

- Line 177: **"Hydrothermal alteration of the shallow lithospheric mantle (<15 km beneath the sea surface) may be suggested, given the anomalously slow sub-crustal P-wave velocities down to at least 15 km"** I am finding it difficult to interpret Figure 2a showing the P-wave velocity. Even as someone who models crustal velocity structure, it seems like this P-wave model is only a little bit slower than typical crust at ~8 km below the surface (sub-crustal depths?) – with no reference, it's difficult to be sure of what the velocities mean in here. Also where is this model exactly relative to the Chain TF? Does it extend past the ridge – to the east of the fault shown in b and c panels? Although I think it is useful to have a velocity model if the authors are arguing for the impact of hydrothermal alteration, in this form, I am not sure I gain much in the way of evidence for hydrothermal processes from this figure.

We have tempered our interpretation of the active source study and provided additional information to support our interpretation. We also added Supplementary material Figure 7, which shows the Moho reflections, which are relatively simple and clear, illuminating a flat and simple structure. For this reason, we interpret the velocity variations beneath the Moho as related to altered mantle material.

We added the following text to the methods: "Our P-wave refraction study illuminates anomalous crust and mantle structure beneath ELFS. There is a clear Moho reflector visible across the region, which is relatively flat at 10 km depth, especially beneath ELFS (Supplementary Material Fig 7, Fig 3a). Within the crust of ELFS there is evidence of a high velocity core, while the crustal structure away from ELFS has relatively flat velocity contours (Fig. 3 a). In the upper most mantle beneath the Moho reflector we observe a broad region of slower than expected mantle velocities (< 7.5 km/s), i.e., 4 – 10 % slower in comparison to velocities >7.8 km/s on the western side of the ELFS. The slow velocity anomaly extends to at least 15 km depth and is centered just east of ELFS."

We added the following text to the main text: "Observed slow P-wave could be explained by thickened crust, enhanced porosity, and/or composition including hydrothermal alteration, which we will discuss in greater detail in the next section."

and

"Clear Moho reflections are visible across the region from the active source experiment, and these arrivals support relatively constant crustal thickness at 11 km beneath the sea surface (Supplementary material, Fig. ED7); however, anomalously slow P-wave velocities in the subcrustal lithosphere are present down to at least 15 km depth beneath the sea surface, which are consistent with fluids and damaged/porous altered low density mantle material."

Supplementary material Figure 5 shows the location of the active source study.

- Line 180: **"Alteration caused by fluid infiltration may also explain why the section containing the longest (50 km) fault scarp in the region is not associated with the largest earthquakes, but instead associated with elevated b-values (1.1 – 1.2) with very low amounts of energy released seismically (4 %)."** This is quite confusing. Are the authors saying here that hydrothermal alteration is prevalent everywhere along the fault – in the asperity and barrier zones? Where specifically are they referring to when they reference the 50 km long fault scarp (is this within the AX or BX zones?).

We amended the section to clarify which fault we are referring to:

"Fluids may also reduce the frictional properties of the TF beneath ELFS and explain why the longest (50 km) fault scarp in the region (Fig. 1, green line on ELFS, Supplementary Material Fig. 1) is not associated with the largest earthquakes, but instead associated with elevated b-values (1.1 – 1.2) with very low fractions of seismically released moment (4 %)."

- Re Methods - S-wave model translated into thermal model: I am not an expert in using S-wave velocities

to estimate thermal structure, but it does seem to me that there could be physical factors other than just temperature that are influencing S-wave velocity at relatively shallow depths, such as the 5-20 km below the seafloor depths that are shown in the paper overlapping with seismicity.

Please see previous response regarding the possible effects of Earth properties on S-wave velocities.

REVIEWERS' COMMENTS

Reviewer #1 (Remarks to the Author):

I am satisfied with how my comments from the previous version of the manuscript have been addressed. In particular, I appreciate the additional statistical testing that was done, and the updated labeling of the figures. I believe this manuscript is now acceptable for publication. I only note one very minor thing that should be corrected: on l. 388, "<-3mgal" should be ">-3mgal").

Reviewer #2 (Remarks to the Author):

Thank you for the detailed response to both my comments and those of the other reviewers. My concerns and questions have been addressed by the authors, and I now recommend this manuscript for acceptance.

My last two remarks are:

- rereading this manuscript for the first time since the original submission, there are quite a few acronyms that are difficult to quickly process. Any reduction of acronyms (such as the simple TF for transform fault) would be greatly appreciated and would improve the readability
- a tracked changes version of the manuscript would have made the understand what was done in revision much easier...!

Reviewer #4 (Remarks to the Author):

This paper looks into the factors that controls the rupture length of large earthquakes on oceanic transform faults. They focused on the Chain Fracture Zone in the mid Atlantic, where the PI-Lab project collected OBS data. They presented correlation between several geophysical and geological observations. They observed that morphological transpressional features is correlated with rupture barriers and proposed that fluids going down through the flower structures and altered the fault.

The paper is well written. The data is mostly supporting the conclusions. I suggest publication with minor revisions. Below are my comments.

Line 28: Why use TFs? OTFs is usually used for oceanic transform faults.

Line 68: The four topographic highs are difficult to find in Figure 1.

Line 198-200: This seems out of place. Or there is not enough information.

Line 265: "model is" is duplicated.

Line 543: The most critical part of this paper is the high resolution bathymetry data. It was not mentioned in the Data availability. I am not familiar with all the rules but shouldn't it be provided for people to verify the geological structures in Figure 1a.

Figure 1: The seismic segment and the flower structure do not aligned well. Part of the WFS2 is within A1. Isn't this contradict to the conclusion?

Figure 2 is very hard to follow. (a) is showing a different section as (b) and (c). It should be pointed out in the caption in the beginning and also provide an explanation. The P-wave model seems to be based on active source survey and only covers the east segment. This information is in the Method section but should be mentioned here. This will clarify things and help readers to understand the figure.

An additional supplemental figure in map view might be helpful. What is the horizontal location

uncertainty of the OBS data? In a map view figure, do all the events follow the main fault trace? Or do any occur on the side faults in the flower structure?

Faults do not operate in this simple way. I suggest to make it clear that the main conclusion is based on Chain only and might not work for all. There are many oceanic transform faults that transit from seismic to barrier without any surface structure change. Of course, some does such as Garrett.

REVIEWER COMMENTS

Reviewer #1 (Remarks to the Author):

I am satisfied with how my comments from the previous version of the manuscript have been addressed. In particular, I appreciate the additional statistical testing that was done, and the updated labelling of the figures. I believe this manuscript is now acceptable for publication. I only note one very minor thing that should be corrected: on l. 388, “<-3mgal” should be “>-3mgal”.

We thank the reviewer for comments and corrections. The typo was corrected.

Reviewer #2 (Remarks to the Author):

Thank you for the detailed response to both my comments and those of the other reviewers. My concerns and questions have been addressed by the authors, and I now recommend this manuscript for acceptance.

My last two remarks are:

- rereading this manuscript for the first time since the original submission, there are quite a few acronyms that are difficult to quickly process. Any reduction of acronyms (such as the simple TF for transform fault) would be greatly appreciated and would improve the readability
- a tracked changes version of the manuscript would have made the understand what was done in revision much easier...!

We thank the reviewer for all comments and suggestions. We replaced TF with transform fault throughout the manuscript. We decided however to keep the abbreviations for ELFS (East Large Flower Structure), WFS (western Flower Structures) and rMBA (residual Mantle Bouguer Anomaly), which are all shown in figure 1 and are explained in the text as well as in the figure captions.

Reviewer #4 (Remarks to the Author):

This paper looks into the factors that controls the rupture length of large earthquakes on oceanic transform faults. They focused on the Chain Fracture Zone in the mid Atlantic, where the PI-Lab project collected OBS data. They presented correlation between several geophysical and geological observations. They observed that morphological transpressional features is correlated with rupture barriers and proposed that fluids going down through the flower structures and altered the fault.

The paper is well written. The data is mostly supporting the conclusions. I suggest publication with minor revisions. Below are my comments.

Line 28: Why use TFs? OTFs is usually used for oceanic transform faults.

We removed all abbreviation of TF from the manuscript.

Line 68: The four topographic highs are difficult to find in Figure 1.

It was very hard to find an appropriate colour scale for the topographic highs to be visible, and at the same time keep the rest of the regional features visible. For that reason, we highlight these areas

within the yellow curves in Fig. 1. In addition, we moved previous Supplementary Figure 1 into the main text (new Fig. 2), which provides a larger, zoomed in version in which the topography can be seen.

Line 198-200: This seems out of place. Or there is not enough information.
We thank the reviewer for pointing this out. We removed this sentence from the manuscript.

Line 265: "model is" is duplicated.
Duplicate removed.

Line 543: The most critical part of this paper is the high resolution bathymetry data. It was not mentioned in the Data availability. I am not familiar with all the rules but shouldn't it be provided for people to verify the geological structures in Figure 1a.
We added all resources in the Data Availability section.

Figure 1: The seismic segment and the flower structure do not aligned well. Part of the WFS2 is within A1. Isn't this contradict to the conclusion?

This is discussed in lines 258 – 271 of the revised (annotated) manuscript:

"The WFSs observations share some similarities to those of ELFS, but also some differences. Similarities include high topography, broad deformation zones, and relatively high S-wave velocities and likely cooler temperatures on average which could be consistent with larger degrees of hydrothermal circulation in the region. The rMBA in the WFSs region changes from negative to positive from east to west. The higher values are likely caused by normal to thinner crustal thickness, which likely dominates the gravity signature²⁵. The large historical earthquakes beneath WFS2 also distinguish it from the other flower structures. One explanation may be that the long fault segment in the region extends 20 km east of the flower structure and may behave more similarly to the long fault in section A2. Another explanation could be that there are temporal variations in rheological behaviour as water is delivered and transported through the system. Episodic fluid delivery and transport and multi-mode rupture models have been previously proposed to explain observed seismicity patterns^{5,7,40} and have also been predicted by geodynamic modelling²⁴ and inferred from samples from exhumed oceanic transform faults⁴¹."

Figure 2 is very hard to follow. (a) is showing a different section as (b) and (c). It should be pointed out in the caption in the beginning and also provide an explanation. The P-wave model seems to be based on active source survey and only covers the east segment. This information is in the Method section but should be mentioned here. This will clarify things and help readers to understand the figure.

We added reviewer's recommendations in the caption of our new figure 2 (previously Supplementary Figure 1).

An additional supplemental figure in map view might be helpful. What is the horizontal location uncertainty of the OBS data? In a map view figure, do all the events follow the main fault trace? Or do any occur on the side faults in the flower structure?

This is now shown in the new Fig. 1 (previously Supplementary Figure 1), where we added the location of the OBS seismicity with the median and 95% quantile of the horizontal uncertainty.

Faults do not operate in this simple way. I suggest to make it clear that the main conclusion is based on Chain only and might not work for all. There are many oceanic transform faults that transit from seismic to barrier without any surface structure change. Of course, some does such as Garrett.

We added the following phase (Lines 287-289 of the revised manuscript) to be more specific: "This also likely occurs elsewhere, particularly in transpressional environments, for example, St. Paul and Owen Transform Faults^{43,44}". (*Maia et al., 2016; Janin et al., 2023*). We also changed some other instances of "transforms" to "the transform."